# Validation of WRF-Chem Model and CAMS Performance in Estimating Near-Surface Atmospheric $CO_2$ Mixing Ratio in the Area of Saint Petersburg (Russia)

**Georgy Nerobelov** [1,*] **, Yuri Timofeyev** [1]**, Sergei Smyshlyaev** [2]**, Stefani Foka** [1]**, Ivan Mammarella** [3] **and Yana Virolainen** [1]

1 Department of Atmospheric Physics, Faculty of Physics, St. Petersburg State University, Universitetskaya emb. 7/9, 199034 Saint-Petersburg, Russia; y.timofeev@spbu.ru (Y.T.); stesy16@mail.ru (S.F.); yana.virolainen@spbu.ru (Y.V.)
2 Department of Meteorological Forecasts, Meteorological Faculty, Russian State Hydrometeorological University, Voronezhskaya St. 79, 192007 Saint-Petersburg, Russia; smyshl@rshu.ru
3 Institute for Atmospheric and Earth System Research/Physics, University of Helsinki, FI 00014 Helsinki, Finland; ivan.mammarella@helsinki.fi
* Correspondence: akulishe95@mail.ru

**Abstract:** Nowadays, different approaches for $CO_2$ anthropogenic emission estimation are applied to control agreements on greenhouse gas reduction. Some methods are based on the inverse modelling of emissions using various measurements and the results of numerical chemistry transport models (CTMs). Since the accuracy and precision of CTMs largely determine errors in the approaches for emission estimation, it is crucial to validate the performance of such models through observations. In the current study, the near-surface $CO_2$ mixing ratio simulated by the CTM Weather Research and Forecasting—Chemistry (WRF-Chem) at a high spatial resolution (3 km) using three different sets of $CO_2$ fluxes (anthropogenic + biogenic fluxes, time-varying and constant anthropogenic emissions) and from Copernicus Atmosphere Monitoring Service (CAMS) datasets have been validated using in situ observations near the Saint Petersburg megacity (Russia) in March and April 2019. It was found that CAMS reanalysis data with a low spatial resolution ($1.9° \times 3.8°$) can match the observations better than CAMS analysis data with a high resolution ($0.15° \times 0.15°$). The CAMS analysis significantly overestimates the observed near-surface $CO_2$ mixing ratio in Peterhof in March and April 2019 (by more than 10 ppm). The best match for the CAMS reanalysis and observations was observed in March, when the wind was predominantly opposite to the Saint Petersburg urbanized area. In contrast, the CAMS analysis fits the observed trend of the mixing ratio variation in April better than the reanalysis with the wind directions from the Saint Petersburg urban zone. Generally, the WRF-Chem predicts the observed temporal variations in the near-surface $CO_2$ reasonably well (mean bias ≈ $(-0.3) - (-0.9)$ ppm, RMSD ≈ 8.7 ppm, correlation coefficient ≈ $0.61 \pm 0.04$). The WRF-Chem data where anthropogenic and biogenic fluxes were used match the observations a bit better than the WRF-Chem data without biogenic fluxes. The diurnal time variation in the anthropogenic emissions influenced the WRF-Chem data insignificantly. However, in general, the data of all three WRF-Chem model runs give almost the same $CO_2$ temporal variation in Peterhof in March and April 2019. This could be related to the late start of the growing season, which influences biogenic $CO_2$ fluxes, inaccuracies in the estimation of the biogenic fluxes, and the simplified time variation pattern of the $CO_2$ anthropogenic emissions.

**Keywords:** $CO_2$ transport modelling; WRF-Chem and CAMS validation; surface mixing ratio; measurements

## 1. Introduction

The gas composition of the atmosphere is essential for different physical, chemical, and biophysical processes on Earth. The growth of the content of greenhouse gases (GHGs) has

changed the radiation balance of the planet, causing an increase in air temperature in the troposphere at a remarkably high rate in the subpolar and polar regions [1]. Anthropogenic emissions of gaseous pollutants significantly impact the environmental conditions in the megacities and industrial regions of many countries. The mentioned consequences have triggered the development of a complex monitoring system (which includes ground-based, aircraft, satellite, and other types of observations) of the spatio-temporal variation in different gases. Various measurements and numerical models are used to study the global cycles of atmospheric gases that are important for the climate and ecology. In addition, high-resolution measurements and modelling can be used to estimate anthropogenic and biogenic sources and sinks of some chemical species on a city scale. This is very important, since megacities have essentially determined ~70% of the anthropogenic carbon dioxide ($CO_2$) emissions (which is the most important anthropogenic GHG) in the last few decades [2,3]. Different approaches for estimating the anthropogenic emissions of $CO_2$ and other GHGs are applied in many countries to ensure their compliance with treaties concerning the reduction in GHGs emissions. For example, the inventory approach is the widely used method for emission estimation, and it is based on information about particular gas emissions from all potential sources (e.g., the amount of fossil fuel burned [4]). However, the errors of this method can vary significantly to up to more than 50% [5] (depending on the country's level of development, the accuracy of emission estimation, the required spatial resolution, etc.).

The necessity of the estimation of GHG emissions using independent and more informative approaches has encouraged scientists to develop various measurement-based techniques [6,7]. Moreover, new sources of measurements such as satellite observations with a high spatial resolution have originated in the last few decades. Satellite measurements of the content of $CO_2$ and other gases have been carried out using different remote methods and instruments (SCIAMACHI, AIRS, TES, IASI, GOSAT, OCO-2, etc.); see, for example, [7]. In satellite-based approaches for the estimation of GHG anthropogenic emissions, first, the inverse problem of atmospheric optics is solved to obtain the spatio-temporal variations in the gas content from measurements of outgoing Earth radiation. Secondly, the inverse problem of atmospheric transport is solved to estimate the surface emissions using the information obtained on changes in the gases over space and time. This approach is known as inverse modelling. The quality of the solution for the atmospheric transport inverse modelling on the city scale depends on (1) measurement errors, which have to be small in order to register the enhancement of $CO_2$ content within the territory of a city; (2) the excellence of the chemistry transport models (CTMs), which simulate the spatio-temporal variability of various chemical species in the atmosphere [8]; (3) the a priori information used. The a priori information consists of the spatio-temporal variation in fluxes (e.g., anthropogenic and biogenic $CO_2$ fluxes) and the initial and boundary conditions (meteorological and chemical).

Examples of the usage of satellite data to estimate the emissions of $CO_2$ and other gases on large scales can be found in [9–22]. These studies demonstrate that the quality of a priori information and, in particular, the quality of CTMs impact greatly on the accuracy of the emission estimation. In a study [23], eleven different techniques of inverse modelling, different atmospheric parameters, CTMs, and a priori information were compared. A comparison of the four-year (2000–2004) mean fluxes on the global scale obtained by different inversions shows the consistency between the fluxes over ocean and land in the north and south (standard deviation $\approx 0.5$ PgC/y) and the larger spread between the inversion data in the tropics (standard deviation $\approx 0.9$ PgC/y). In the case of estimates on a regional scale, the fluxes calculated using different transport models and a priori data vary from 5 to 96% for particular regions (e.g., 49% for Europe).

However, emission estimation on a city scale is also crucial for the validation of emission inventories for specific cities. In addition, to control the emissions from the territory of a city, it is also important not only to estimate the integral emissions but also the fluxes from specific city areas (for instance, see the study [24]). In the last several years, plenty of studies

on the inverse modelling of anthropogenic $CO_2$ emissions from the territories of large cities and megacities have been carried out. For example, in studies [25–28], perspectives on using the differential column technique to estimate anthropogenic emissions are presented based on ground-based accurate remote measurements and chemistry transport modelling. Examples of the application of inverse modelling on a city scale based on accurate remote measurements (ground-based and satellite) and high-resolution transport modelling can be found in studies [29–32].

Hence, procedures to validate the accuracy of atmospheric transport models and especially their adaptation to the area of research are very important in the inverse modelling of anthropogenic GHG emissions on the city scale. In particular, the validation is important for modelling data of a high spatial resolution (several kilometers), since in this case, the processes for smaller spatial and time scales can influence the model performance. In addition, it is important to assess the quality of the a priori data used in the modelling.

In the current study, we validate the performance of the numerical modelling of $CO_2$ transport in a surface layer near the Saint Petersburg megacity (Russia) during March and April 2019 provided by the Weather Research and Forecasting—Chemistry (WRF-Chem) regional model with a high spatial resolution (3 km) and Copernicus Atmosphere Monitoring Service (CAMS) data. The CAMS data (forecast, analysis, and reanalysis) are used in different operational applications, such as, for example, numerical modelling, where the data can be applied as initial and boundary conditions. In this study, we provide a comparison of two CAMS products—global reanalysis and analysis. Notably, the CAMS analysis data were used as the initial and boundary conditions in our WRF-Chem simulations.

The descriptions of the measurements, WRF-Chem modelling, and CAMS data are given in Section 2, Section 3, and Section 4, respectively; the validation of the wind parameters, CAMS, and WRF-Chem data with respect to the observations is presented in Section 5; the main conclusions of this study along with suggestions for future research are provided in the last section, Section 6.

## 2. $CO_2$ Measurements and the Area of Interest

Continuous in situ measurements of surface atmospheric $CO_2$ mixing ratio and other gases were performed from January 2013 in the building of the Faculty of Physics of Saint Petersburg State University in Peterhof (59.88° N, 29.83° E, Saint Petersburg, white circle on Figure 1) using the Los Gatos Research Greenhouse Gas Analyzer (LGR GGA-24-r-EP), which was set up approximately 30 m above sea level [33]. The precision of the measurements was in the range 50–150 ppb depending on the accumulation time of the instrument (100–5 s, respectively). The GGA is calibrated at least once per week. The frequency of the initial observation data is approximately 20 s. There are many different approaches used to average large observation datasets. In our study, we used a method that required the medians to be calculated first. The medians were found for every 15-minute term for data ranging from 15 min before to 15 min after the term. The medians removed some local extreme values in the temporal variation in the observed $CO_2$ mixing ratio, which could not be simulated by WRF-Chem modelling at a 3 km spatial resolution. We also examined several different "configurations" of the averaging method, varying the period of the averaging and medians. We found that there were no significant differences between the results of the methods. To validate the modelled data, we calculated the average atmospheric $CO_2$ mixing ratio from the estimated medians for time ranging from 1 h before and to 1 h after every 1 h term (to compare with the WRF-Chem data) and 3 h before and 3 h after every 6 h term (to compare with the CAMS data). We assume that the average observations could characterize the $CO_2$ mixing ratio for relatively extended air masses (up to dozens of kilometers, providing that the wind speed is 2–3 m/s).

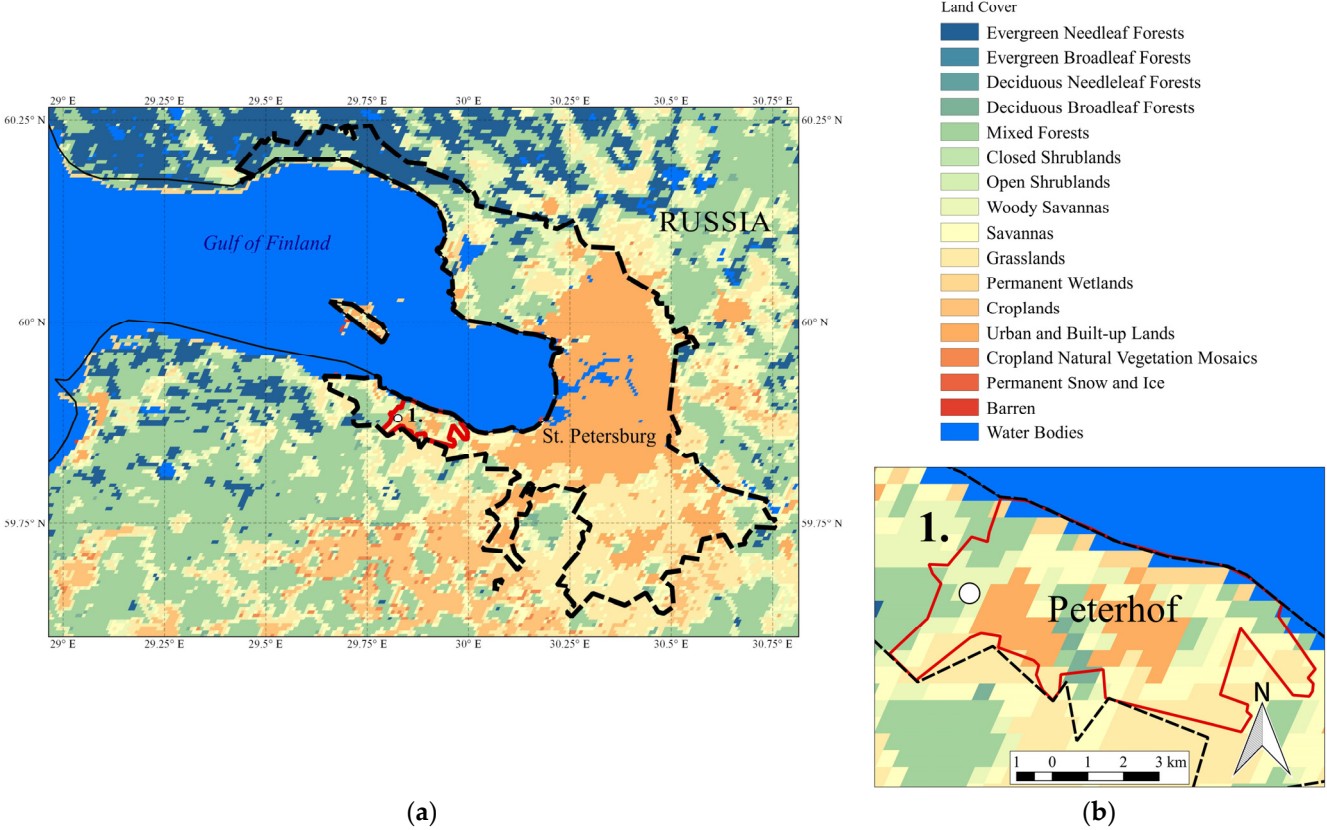

**Figure 1.** Land types according to the Annual International Geosphere–Biosphere Programme (IGBP) classification derived from MODIS (MCD12Q1 v006) for the territory of Saint Petersburg (**a**) and Peterhof (**b**) (Russia); the white circle depicts the position of the Peterhof measurement station.

Peterhof (Figure 1, red solid line) is a suburb of the Saint Petersburg megacity (Figure 1, black dotted line), which is located in a green area with a limited roadway network. According to the Moderate Resolution Imaging Spectroradiometer (MODIS) satellite data on land types (satellites Aura and Terra, product MCD12Q1 v006, https://lpdaac.usgs.gov/products/mcd12q1v006/ (accessed on 15 November 2019)), approximately 25% of the Peterhof territory is occupied by urban and built-up land, when the other part of the city is occupied by different plants, varying from grasslands to mixed forests. The city is surrounded by a continuous water surface from the north (the Gulf of Finland) and mainly grasslands and mixed forests on the other sides. The vegetation surrounding could potentially be a large source of biogenic $CO_2$ flux during the growing season. The urbanized area of the megacity Saint Petersburg is located east of Peterhof at an approximately 40 km distance.

The spatial distribution of anthropogenic $CO_2$ emissions with a high spatial resolution (1 km) from the Open-Source Data Inventory for Anthropogenic $CO_2$ (ODIAC) dataset [34] for 2018 in Saint Petersburg and its surroundings is given in Figure 2. In addition, the monthly averaged wind directions at a 10 m height according to the meteorological reanalysis ERA5 data [35] for March (red arrow) and April (blue arrow) of 2019 are presented in Figure 2. Dotted arrows represent the standard deviation of the wind direction data, characterizing the direction variability during both months. In Figure 2, the wind direction arrows show where the air masses are moving. According to these data, the prevailing wind directions and their average variations were approximately $226 \pm 67°$ and $162 \pm 106°$, respectively, in March and April 2019. It is easy to notice that the areas with the highest $CO_2$ emissions (Figure 2) coincide well with the positions of urbanized territories according to the MODIS data (Figure 1). It also can be seen that the values of the emissions from Saint Petersburg city center exceed by several times the emissions from the sources that

surround Peterhof. This points to the fact that variation in wind speed and direction can significantly determine the ground-level $CO_2$ mixing ratio on the territory of Peterhof. In agreement with the provided $CO_2$ anthropogenic emissions and wind direction data, one can assume that the average ground-level $CO_2$ mixing ratio in March is lower than that in April 2019, while the trend of the mixing ratio variation may be more homogeneous in March.

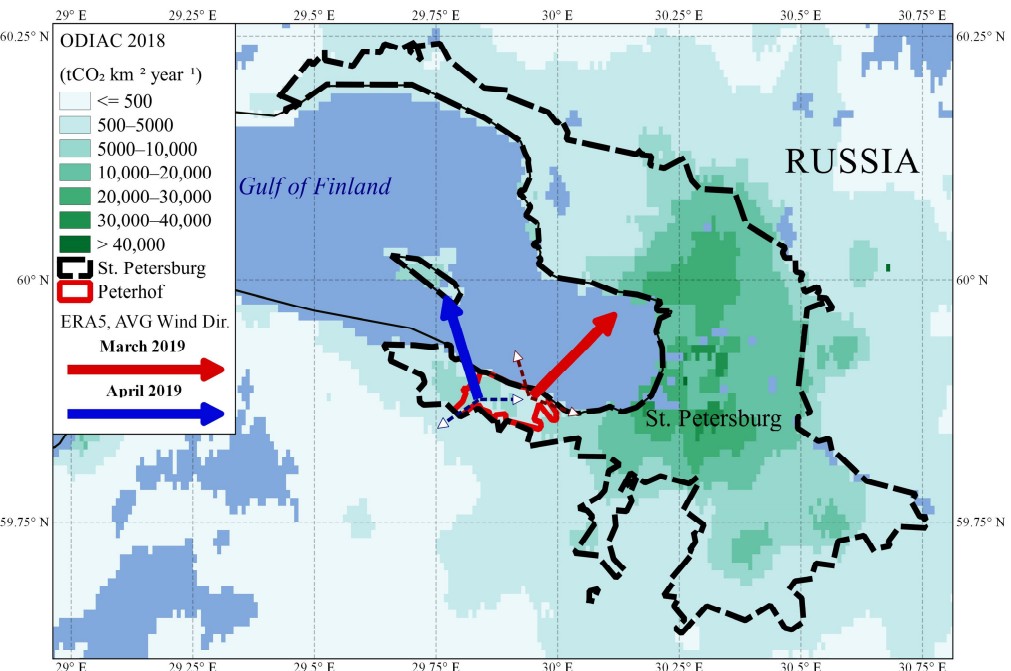

**Figure 2.** Anthropogenic $CO_2$ emissions from the high spatial resolution ODIAC (2018) and monthly averaged (March and April 2019) wind direction at 10 m and its standard deviation (dotted arrows) for the territory of Saint Petersburg and Peterhof (Russia).

## 3. WRF-Chem Modelling of $CO_2$ Spatio-Temporal Variation

The numerical weather prediction and chemistry transport model WRF-Chem (Weather Research and Forecasting—Chemistry) version 4.1.2 [36–38] was used in this research. The modelling system is able to perform the simulation of transport, mixing, and chemical transformations of gases and aerosols in an online mode (simultaneous calculation of meteorological and chemical parameters). The transport of long-lived gases (e.g., $CO_2$, $CH_4$) can be treated in a "passive" way—i.e., without chemical transformations. In the current study the WRF-Chem model was used to simulate spatio-temporal distribution of $CO_2$ using three scenarios of prior $CO_2$ sources and sinks—time-varying anthropogenic and biogenic fluxes, time-varying anthropogenic and constant anthropogenic emissions.

The modelling was performed for March and April 2019. We focused on this period due to the availability of in situ and remote measurements for the territory of Saint Petersburg and its suburbs. However, only a comparison with in situ observations is provided in this study due to the fact that the comparison of the WRF-Chem data with the remote observations is still in progress. Nesting mode was applied in the modelling with 9 and 3 km spatial resolutions for a parent (outer) and daughter (inner) domains, respectively. The coverage of the domains can be seen in Figure 3.

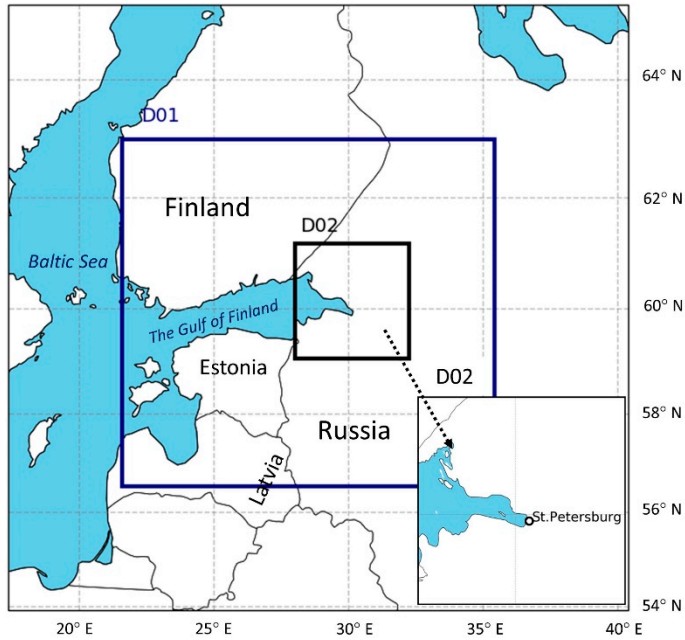

**Figure 3.** WRF-Chem modelling domains.

The outer domain (D01) covers the Leningrad region, the Gulf of Finland, the southern part of Finland, Estonia, and part of Latvia. The inner domain (D02) contains more than half of the Leningrad region territory, with Saint Petersburg in the center. Six model runs were performed. In runs 1ab, we considered biogenic and anthropogenic $CO_2$ sources and sinks, which vary in time; in runs 2ab and 3ab, only varying and constant in time anthropogenic emissions were taken into account, respectively (Table 1). In this research, we used 39 vertical hybrid layers from the surface up to 50 hPa. The main characteristics of the WRF-Chem simulations can be found in Table 1. The WRF-Chem physics configuration for all the model runs as well as for the modelling domains was the same and is the following: microphysics: WRF single-moment 3-class scheme; radiation: Rapid and accurate Radiative Transfer Model (RRTM) longwave scheme and Goddard shortwave scheme; cumulus parametrization: Grell 3D ensemble scheme; Planetary Boundary Layer (PBL) scheme: Mellor–Yamada Nakanishi Niino (MYNN) level 2.5 scheme; land surface: Unified Noah Land Surface Model.

**Table 1.** The main characteristics of the Weather Research and Forecasting—Chemistry (WRF-Chem) runs.

| No of WRF-Chem Model Run | | 1a | 1b | 2a | 2b | 3a | 3b |
|---|---|---|---|---|---|---|---|
| **Horizontal resolution** | | D01—9 km, D02—3 km | | | | | |
| **Vertical resolution** | | 39 hybrid vertical layers (up to 50 hPa) | | | | | |
| **Initial and boundary conditions** | **Meteorology** | GFS ANL (0.5°, 3 h) | | | | | |
| | **Atmospheric $CO_2$ mixing ratio** | CAMS Global analysis of $CO_2$ (0.15°, 6 h) | | | | | |
| **Length of simulation** | | March 2019 | April 2019 | March 2019 | April 2019 | March 2019 | April 2019 |
| **$CO_2$ sources and sinks** | **Anthropogenic sources** | ODIAC 2018, diurnal temporal variation | | | | ODIAC 2018, no temporal variation | |
| | **Biogenic sources and sinks** | VPRM, temporal variation—3 h | | No biogenic fluxes | | No biogenic fluxes | |

### 3.1. Initial and Boundary Conditions

Global forecast system (GFS) analysis with a 0.5° horizontal resolution on 64 hybrid vertical levels (approximately up to 55 km) and with a 3 h frequency was used as the meteorological initial and boundary conditions [39]. To implement meteorological data for the WRF-Chem modelling, we used a set of preprocessors, which were developed for the WRF models: WRF preprocessing system (WPS). The WPS consists of three main programs—geogrid (creates a modelling domain with geophysical parameters such as landscape, soil temperature, land type, etc.), ungrib (transforms meteorological input data to a special format), and metgrid (interpolates meteorological data, transformed by ungrib, to the WRF modelling domain created by geogrid). To set the atmospheric $CO_2$ initial and boundary conditions, the data of the Copernicus Atmosphere Monitoring Service (CAMS) were used [40]. The service provides an analysis of the $CO_2$ spatio-temporal variation on a global scale with a 0.15° horizontal resolution on 137 hybrid vertical levels (approximately up to 80 km) every 6 h. The data were simulated by a global atmospheric circulation model integrated forecast system (IFS). To set up the chemical initial and boundary conditions, we used the "mozbc" tool, which was provided by the Atmospheric Chemistry Observations and Modeling Lab (ACOM) of National Center for Atmospheric Research (NCAR) [41].

### 3.2. $CO_2$ Sources and Sinks

We used in our simulations a $CO_2$ anthropogenic emission inventory with a high spatial resolution ($\approx$1 km) ODIAC [34] to set the anthropogenic sources. The dataset consists of monthly varying sums of $CO_2$ emissions from different manmade sources (except for international aviation and shipping). Since the ODIAC data are a sum of all the $CO_2$ anthropogenic emissions for a month of a specific year, the original anthropogenic emissions were constant during the period of simulation. According to the study [42], we applied time factors for the 1ab and 2ab model runs, which allowed the data to vary hourly during the day. Hence, unitary diurnal variation was implemented for every simulation day, which means that there was no weekly variation. The runs 3ab were performed without the diurnal variability of the anthropogenic emissions. Since the ODIAC dataset was not available for 2019, we used the data from 2018.

We used the vegetation photosynthesis and respiration model (VPRM) [43] to calculate the biogenic $CO_2$ fluxes at a 3 h frequency. These fluxes are the result of the difference in $CO_2$, which is released and absorbed by vegetation. That means that the biogenic fluxes can have a positive (source) or negative (sink) sign. Data such as satellite measurements (Earth surface reflectance in visible and infrared electromagnetic spectra), meteorological observations (air temperature), information on photosynthetically active radiation, information on vegetation type, and some specific a priori-defined constants are needed to calculate $CO_2$ biogenic fluxes. Since we could not operate an existing VPRM preprocessor [38,44], we wrote a simple VPRM program, which conserves all principles from its original description in [43] and calculates $CO_2$ biogenic emissions with a 0.25° spatial resolution. To validate the estimated biogenic fluxes, the experimental eddy covariance $CO_2$ flux data collected at the Station for Measuring Forest Ecosystem–Atmosphere Relations (SMEAR II) located in Hyytiälä (61.85° N and 24.28° E), southern Finland, were used. The SMEARII flux tower is located in a 57-year-old (in 2019) Scots pine (Pinus sylvestris L.) stand, which is homogeneous for about 200 m in all directions and maximally extends to the north about 1 km [45]. The instrument system was set up at a height of 27 m (tall tower) and consisted of a Gill HS-50 three-dimensional ultrasonic anemometer measuring three wind velocity components, sonic temperature, and a LI-COR LI-7200 gas analyzer for a specific $CO_2$ and $H_2O$ mixing ratio. The $CO_2$ fluxes were calculated using the state-of-the-art methodologies [46,47].

The temporal variations in the 3-hourly averaged modelled and observed biogenic fluxes are given in Figure 4. Here, we present the modelled data from the cell, which is nearest to the observation station (61.8° N, 22.3° E). The correlation coefficients between the modelled and observation data are approximately 0.63. Even though we did not expect a

perfect agreement between the data of the local-scale measurement station and the VPRM, the comparison demonstrates an adequate agreement between the simulated and observed average diurnal variation in the biogenic fluxes.

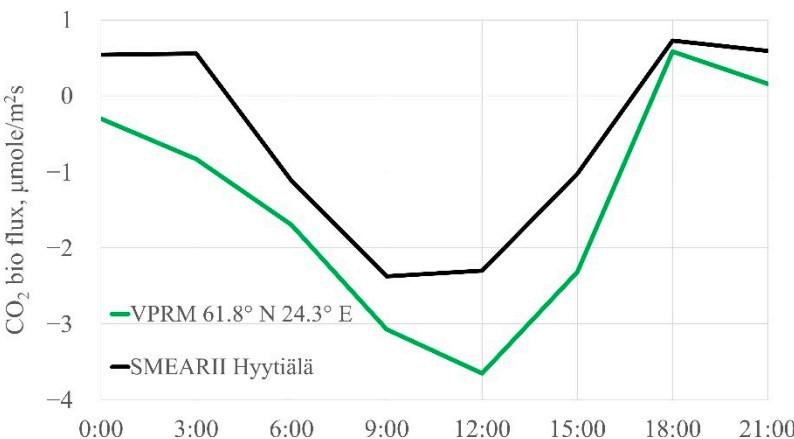

**Figure 4.** Averaged diurnal cycle of $CO_2$ biogenic fluxes according to VPRM and Hyytiälä observations for March–April 2019.

To consider the anthropogenic and biogenic fluxes in the WRF-Chem simulations (1ab), the final $CO_2$ sources and sinks were found simply as a sum of the two types of fluxes. After that, the resulting fluxes were processed using the "anthro_emiss" software, which was provided by the Atmospheric Chemistry Observations and Modeling Lab (ACOM) of the National Center for Atmospheric Research (NCAR).

## 4. CAMS Data of $CO_2$ Spatio-Temporal Variation

Copernicus Atmosphere Monitoring Service (CAMS) provides the forecast, analysis, and reanalysis of the spatio-temporal variation in greenhouse gases on a global scale [48]. In the current research, we used two CAMS products: the reanalysis (v19r1) and analysis of the spatio-temporal variation in $CO_2$ on a global scale. The description of the products can be found here [40,49,50]. The CAMS reanalysis data were provided for a wide time range every 3 h. The horizontal resolution constitutes approximately 1.9° and 3.8°, whereas vertically the data are distributed on 39 hybrid levels from the Earth's surface to higher than 70 km. The data were obtained by the global circulation model Laboratoire de Météorologie Dynamique (LMDz) [51] using biogenic fluxes and anthropogenic emissions and by the assimilation of surface air-sample measurements. The CAMS scientists carry out validation for every new version of the product. According to the validation of the v19r1 product from the report [49], in general, the biases between the CAMS reanalysis data and surface measurements for the period 1979–2020 are less than 1 ppm. The CAMS analysis data were the same as those we used for the initial and boundary conditions in the WRF-Chem modelling (see Table 1 and Section 3.1).

## 5. Results and Discussion

### 5.1. Validation of WRF-Chem Wind Speed and Direction by ERA5 Reanalysis

Wind speed and direction are essential for the spatial–temporal variation in $CO_2$ in the atmosphere. The comparison of these wind parameters obtained by the WRF-Chem model and ERA5 meteorological reanalysis [35] was carried out in our study. The ERA5 reanalysis data are available with an approximately 30 km horizontal resolution on hybrid levels up to about 80 km. The reanalysis and WRF-Chem data were used every 1 h for the period March–April 2019. To obtain the wind speed and direction, we processed meridional (u) and latitudinal (v) wind components at a 10 m height from both datasets using Equations (1) and (2):

$$Wind\ Speed = \sqrt{u^2 + v^2}, \tag{1}$$

$$Wind\ Direction = arctan\frac{v}{u}. \qquad (2)$$

The data for the comparison were taken from the nearest cells. The coordinates of the cell are the following—60.00° N, 29.75° E for the ERA5 and 59.88° N, 29.81° E for the WRF-Chem data. The temporal variation in the wind speed and direction at 10 m and the difference in March–April 2019 for the territory of Peterhof according to the ERA5 and WRF-Chem data are given in Figures 5 and 6. The main trends for the wind speed and direction of the WRF-Chem data fit reasonably well with the ERA5 trends. The maximal discrepancies in the wind speed between the two datasets can be seen in March during the periods when the speed was relatively high or low (for example, 9 and 24 March). By contrast, the highest mismatches in the wind direction were registered in April (see, for example, the period 8–26). A better fit for the wind speed and direction trend according to the WRF-Chem and ERA5 data can be observed in March 2019.

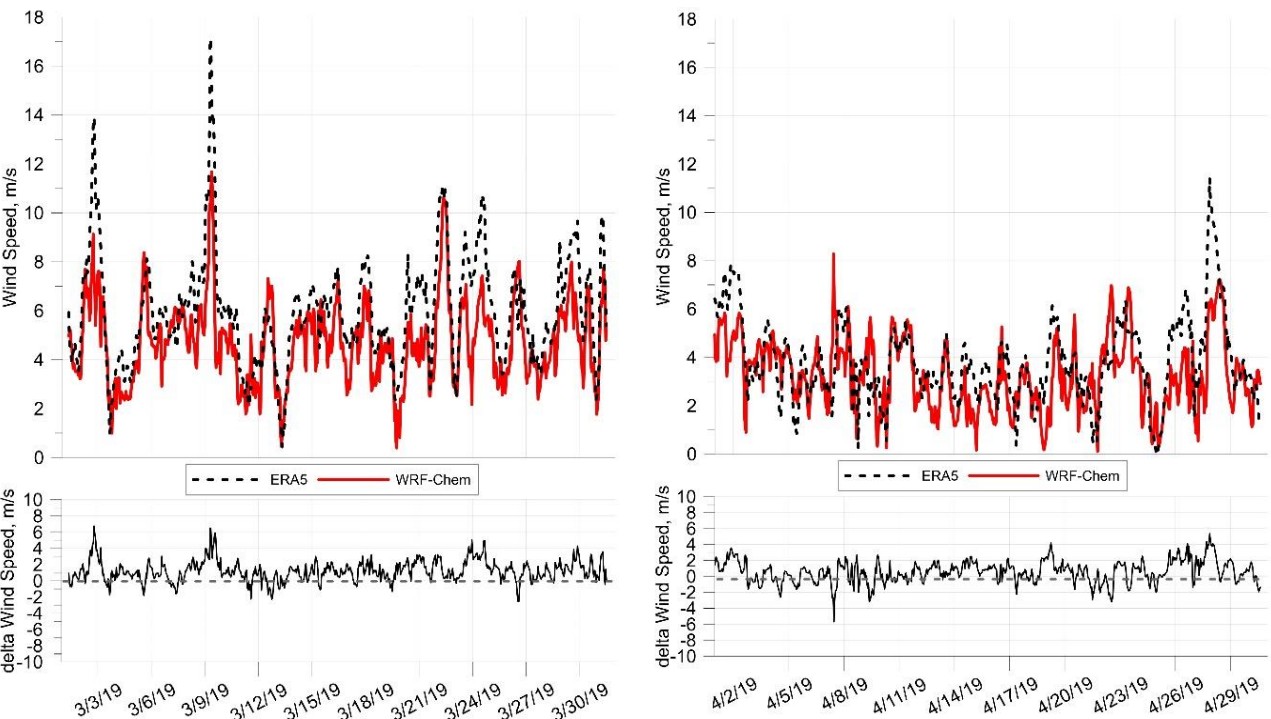

**Figure 5.** Temporal variation in wind speed at 10 m and differences according to the ERA5 and WRF-Chem data in March (**left**) and April (**right**) 2019 in the territory of Peterhof (Saint Petersburg).

The comparison of the two datasets (Table 2) demonstrates that their means and standard deviations (SD shows the variability of the data relative to the mean) for the wind speed are quite similar. However, the mean and SD of the ERA5 data are slightly higher in both months. The best match for the wind speed means and SDs according to both datasets was observed in April—the ERA5 mean and SD is slightly higher (by 0.6 and 0.2 m/s, respectively) than the WRF-Chem parameters. The ERA5 mean wind speed and SD were higher by 1.3 and 0.5 m/s, respectively, in March 2019. However, a better fit of the wind direction according to the ERA5 and WRF-Chem datasets was found in March than in April 2019. The difference between the mean and SD is 4.8° and 3.8° in March and 13.8° and 0.1° in April. Notably, the wind direction variabilities in April 2019 according to the WRF-Chem and ERA5 data are significantly higher than those in March (SDs are 106 and 63–67°, respectively). In both cases, the ERA5 SDs are slightly higher than the WRF-Chem SDs. The best fit trend of the wind parameters according to Figures 5 and 6 and small discrepancies between the ERA5 and WRF-Chem means and SDs can point to a good agreement in the natural variability of both datasets.

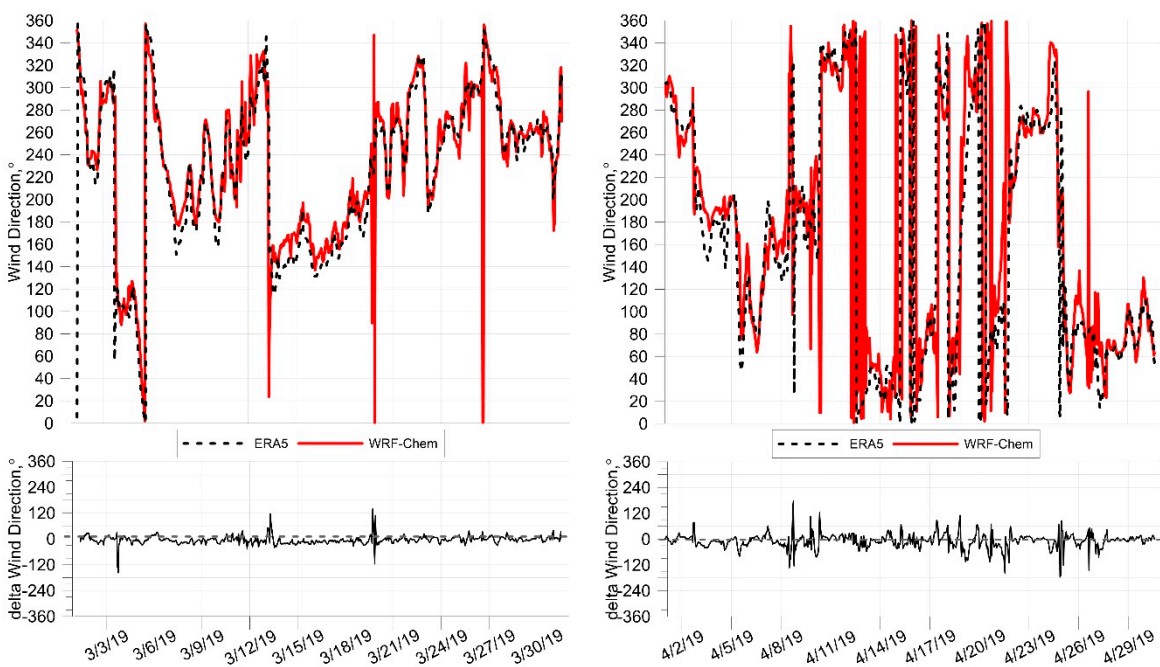

**Figure 6.** Temporal variation in wind direction at 10 m and differences according to the ERA5 and WRF-Chem data in March (**left**) and April (**right**) 2019 in the territory of Peterhof (Saint Petersburg).

**Table 2.** Statistical characteristics of the wind speed and direction at 10 m according to the ERA5 and WRF-Chem data for Peterhof in March and April 2019.

| Wind Speed (m/s) | | | | |
|---|---|---|---|---|
| Period | March 2019 | | April 2019 | |
| | ERA5 | WRF-Chem | ERA5 | WRF-Chem |
| Mean± SD | $6.0 \pm 2.2$ | $4.7 \pm 1.7$ | $3.8 \pm 1.7$ | $3.2 \pm 1.5$ |
| **Wind Direction (°)** | | | | |
| Period | March 2019 | | April 2019 | |
| | ERA5 | WRF-Chem | ERA5 | WRF-Chem |
| Mean± SD | $225.8 \pm 66.8$ | $230.6 \pm 63.0$ | $162.0 \pm 106.0$ | $175.8 \pm 105.9$ |

Table 3 shows the characteristics of the difference between the ERA5 and WRF-Chem wind data. These are the mean bias or M, root-mean-square deviation or RMSD, and correlation coefficient or R.

**Table 3.** Statistical characteristics of the wind speed and direction differences between ERA5 and WRF-Chem data for Peterhof in March and April 2019.

| Wind Speed | | | |
|---|---|---|---|
| Date | M, m/s | RMSD, m/s | R |
| March | 1.2 | 1.8 | $0.82 \pm 0.04$ |
| April | 0.6 | 1.5 | $0.63 \pm 0.06$ |
| March–April | 0.9 | 1.6 | $0.80 \pm 0.03$ |
| **Wind Direction** | | | |
| Date | M, ° | RMSD, ° | R |
| March | $-4.8$ | 18.9 | $0.87 \pm 0.04$ |
| April | $-8.5$ | 35.1 | $0.63 \pm 0.06$ |
| March–April | $-6.6$ | 28.1 | $0.73 \pm 0.04$ |

The mean biases illustrate that the ERA5 wind speed values are higher on average than the WRF-Chem values in both months by approximately 0.9 m/s. Here, the RMSD characterizes how the datasets differ in general and if there are large outliers in the differences. The RMSDs for the wind speed are slightly smaller in April than in March, changing from 1.8 to 1.5 m/s. However, the R is higher in March than in April ($0.82 \pm 0.04$ vs. $0.63 \pm 0.06$). In contrast, the wind direction RMSD appeared to be smaller by almost two times in March ($18.9°$ vs. $35.1°$). The R of the wind speed was also higher in March ($0.87 \pm 0.04$ and $0.63 \pm 0.04$, respectively).

The analysis shows that the wind speed in a surface layer according to the WRF-Chem data in March and April 2019 for the territory of the Saint Petersburg suburb matches well with the ERA5 wind speed. Nevertheless, the RMSDs show that the WRF-Chem wind directions fit with the ERA5 data worse in April than in March 2019. Considering all the mentioned data together, we can conclude that the fit of the wind data was better in March 2019 than in April, which can also be seen from Figures 5 and 6. This could be related to the quality of the initial and boundary meteorological conditions for March and April. Even though we used the same meteorological data product (GFS ANL) for both months, the WRF-Chem simulations for March and April 2019 were initialized and run independently. Another reason for the larger disagreements in April could be the difficulties in the WRF-Chem numerical modelling at a high spatial resolution (3 km), which can be related to a specific meteorological situation.

In the study [25], a comparison of the wind speed and direction at 10 m according to a WRF-GHG simulation and observation data was carried out. The research shows that the RMSDs for the wind speed and direction are approximately 1 m/s and $61°$ for the 1–10 July 2014 period. The wind speed RMSD is close to the one we obtained, but the wind direction RMSD is larger by almost two times.

*5.2. Validation of CAMS Near-Surface CO$_2$ Mixing Ratio*

The observations of the surface atmospheric CO$_2$ mixing ratio in Peterhof allow us to validate two CAMS products: the global reanalysis v19r1 [49,52] and analysis [40] datasets. The second one was used for the initial and boundary conditions in the WRF-Chem modelling. We obtained CAMS data on the lowest model layer from model cells with the coordinates $59.68°$ N, $30.0°$ E (reanalysis) and $59.95°$ N, $29.75°$ E (analysis). Because of the CAMS analysis time resolution, we employed other datasets with a 6 h frequency. The temporal variation in the surface atmospheric CO$_2$ mixing ratio according to the CAMS and observation data in Peterhof in March and April 2019 can be seen in Figure 7.

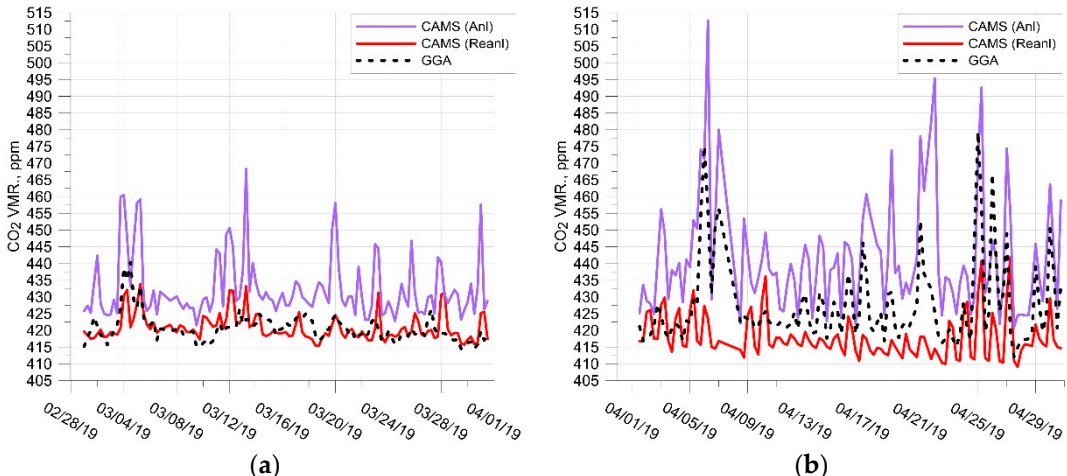

**Figure 7.** Temporal variation in the surface atmospheric CO$_2$ mixing ratio according to the CAMS and in situ observation data in Peterhof in March (**a**) and April (**b**) 2019.

The graphs demonstrate that the CAMS analysis data significantly overestimate the values for the $CO_2$ mixing ratio relative to the CAMS reanalysis and observation data for the suburb of Saint Petersburg. The mismatches are significantly higher in April and sometimes reach more than 50 ppm. The CAMS reanalysis data agree well with the observations in March but also differ significantly in April 2019, underestimating the actual values sometimes by approximately 50 ppm. The reanalysis data have the best match with the observed trend of the $CO_2$ mixing ratio in March, while the analysis data fit it better in April.

The means and standard deviations of the observation and CAMS data are provided in Table 4. The standard deviations, which were estimated for the observation and modelled data relative to their means characterize the "natural variability" of the data and can be compared with each other because the datasets have the same time and spatial resolution. It can be noted that the mean and SD of the observation data almost totally correspond to the data of the CAMS reanalysis, and the difference is equal to 0.2 ppm in March. In contrast, the mean and SD of the CAMS analysis data appear to be higher than those of the observation data by 12 and 5.2 ppm, respectively. However, both the CAMS products possess large mismatches with the observations in April. The mean and RMSD of the CAMS analysis data are higher by 14.4 and 4.5 ppm when the parameters according to the CAMS reanalysis are lower by 9.4 and 6.1 ppm, respectively, relative to the in situ observations.

**Table 4.** Statistical characteristics of the CAMS and observation data for Peterhof in March and April 2019; Reanl: reanalysis; Anl: analysis.

| Period | March 2019 | | | April 2019 | | |
|---|---|---|---|---|---|---|
| | GGA | CAMS (Reanl) | CAMS (Anl) | GGA | CAMS (Reanl) | CAMS (Anl) |
| Mean $\pm$ SD (ppm) | $420.6 \pm 4.2$ | $420.8 \pm 4.0$ | $432.6 \pm 9.4$ | $427.4 \pm 12.6$ | $418.0 \pm 6.5$ | $441.8 \pm 17.1$ |

Table 5 shows the parameters, which characterize the difference between the observation and CAMS data (mean bias or M, root-mean-square deviation or RMSD, and correlation coefficient or R). The best agreement with the observations in March was found for the CAMS reanalysis data. For example, the mean bias M and RMSD constitute $-0.1$ and 4.2 ppm, respectively. In contrast, the M and RMSD of the CAMS analysis data are significantly higher and equal to $-11.8$ and 14.4 ppm, respectively, in March. The reanalysis and analysis data on average overestimate the real ground-level $CO_2$ mixing ratio in this month. The higher RMSD between the CAMS analysis and observation data suggests that there are more chaotic and steep differences in comparison to the reanalysis data. R appeared to be insignificantly higher for the analysis data than for the reanalysis ($0.52 \pm 0.15$ and $0.46 \pm 0.16$). The reanalysis data correspond to the observations significantly worse in April than in March 2019. However, it is still a little better than the analysis (except for the correlation coefficient). On average, the reanalysis data underestimate the actual mixing ratio (9.4 ppm) when the analysis overestimates it ($-14.3$) in April. The RMSD according to the reanalysis is 15.1 ppm, which is approximately 4 ppm lower than that of the analysis RMSD. Nevertheless, R according to the CAMS reanalysis is approximately in two times lower than that according to the CAMS analysis ($0.37 \pm 0.18$ and $0.69 \pm 0.14$). The better match of the $CO_2$ mixing ratio trend between the CAMS analysis and observation data can be clearly seen in Figure 7b.

**Table 5.** Statistical characteristics of the difference between the CAMS and observation data for Peterhof in March and April 2019; Reanl: reanalysis; Anl: analysis.

| Period | March 2019 | | April 2019 | |
|---|---|---|---|---|
| | GGA−CAMS (Reanl) | GGA−CAMS (Anl) | GGA−CAMS (Reanl) | GGA−CAMS (Anl) |
| M, ppm | −0.1 | −11.8 | 9.4 | −14.3 |
| RMSD, ppm | 4.2 | 14.4 | 15.1 | 19.0 |
| R | 0.46 ± 0.16 | 0.52 ± 0.15 | 0.37 ± 0.18 | 0.69 ± 0.14 |

It was illustrated that the CAMS reanalysis data of quite a crude spatial resolution (1.9° × 3.8°) can have more agreements with the observations of the ground-level $CO_2$ mixing ratio than the CAMS analysis data with a high spatial resolution (about 0.15°) for the area of the Saint Petersburg megacity. The better agreement of the CAMS reanalysis data with the observations can be explained by the procedure of the surface observation assimilation, which can handle the appearance of outliers in the modelled data. In addition, the specific meteorological conditions (especially wind speed and direction), the quality of the a priori data (emissions, initial, and boundary conditions), the spatial resolution of the modelling, and the complexity of the model can have a significant influence on the simulation results. According to the ERA5 wind analysis, the wind speed at 10 m on average was higher in March than in April 2019, with the prevailing wind directions varying from 135° to 360° (opposite to those of Saint Petersburg). By contrast, about one third of the wind direction values were in the range from 35° to 135° in April (from the territory of Saint Petersburg). Hence, the quality of the $CO_2$ fluxes used in the CAMS in March could influence the modelled data less than that in April. The more frequent transport of air masses from the territory of Saint Petersburg in April 2019 and the crude spatial resolution probably were the reasons why the CAMS reanalysis gave a poorer representation of the $CO_2$ mixing ratio trend compared to the CAMS analysis in this month.

*5.3. Validation of WRF-Chem Near-Surface $CO_2$ Mixing Ratio*

We started the analysis from the presentation of histograms, which exhibit the variabilities of the WRF-Chem and observation data (Figure 8). The WRF-Chem data where only time-varying and constant anthropogenic emissions were used differ insignificantly, and that is why only one of these datasets is shown in Figure 8 (with time-varying anthropogenic emissions). The figures demonstrate that higher differences between the WRF-Chem and observation data are registered in March than in April 2019. In addition, the observations have a larger variability (wider histograms) in comparison to the modelled data. A similar situation can be seen in April. However, the WRF-Chem and observation data variabilities were higher in this month. The best agreement between the observations and the WRF-Chem data was found for the modelled results where the anthropogenic and biogenic fluxes were used (Figure 8a,c). Table 6, where the statistical characteristics (mean and standard deviation) of each dataset are given, also proves that.

**Table 6.** Statistical characteristics of the observation and WRF-Chem data for Peterhof in March and April 2019. Here, t.v. is temporal variation; t.const. is time constant.

| Period | Average ± SD (ppm) | | | |
|---|---|---|---|---|
| | GGA | WRF-Chem (t.v. anth + bio) | WRF-Chem (t.v. anth) | WRF-Chem (t.const. anth) |
| March 2019 | 420.7 ± 4.5 | 422.4 ± 4.7 | 423.3 ± 4.9 | 423.4 ± 5.1 |
| April 2019 | 427.3 ± 14.3 | 426.1 ± 9.3 | 426.1 ± 8.2 | 426.3 ± 9.0 |
| March–April 2019 | 423.9 ± 10.9 | 424.2 ± 7.5 | 424.7 ± 6.9 | 424.8 ± 7.4 |

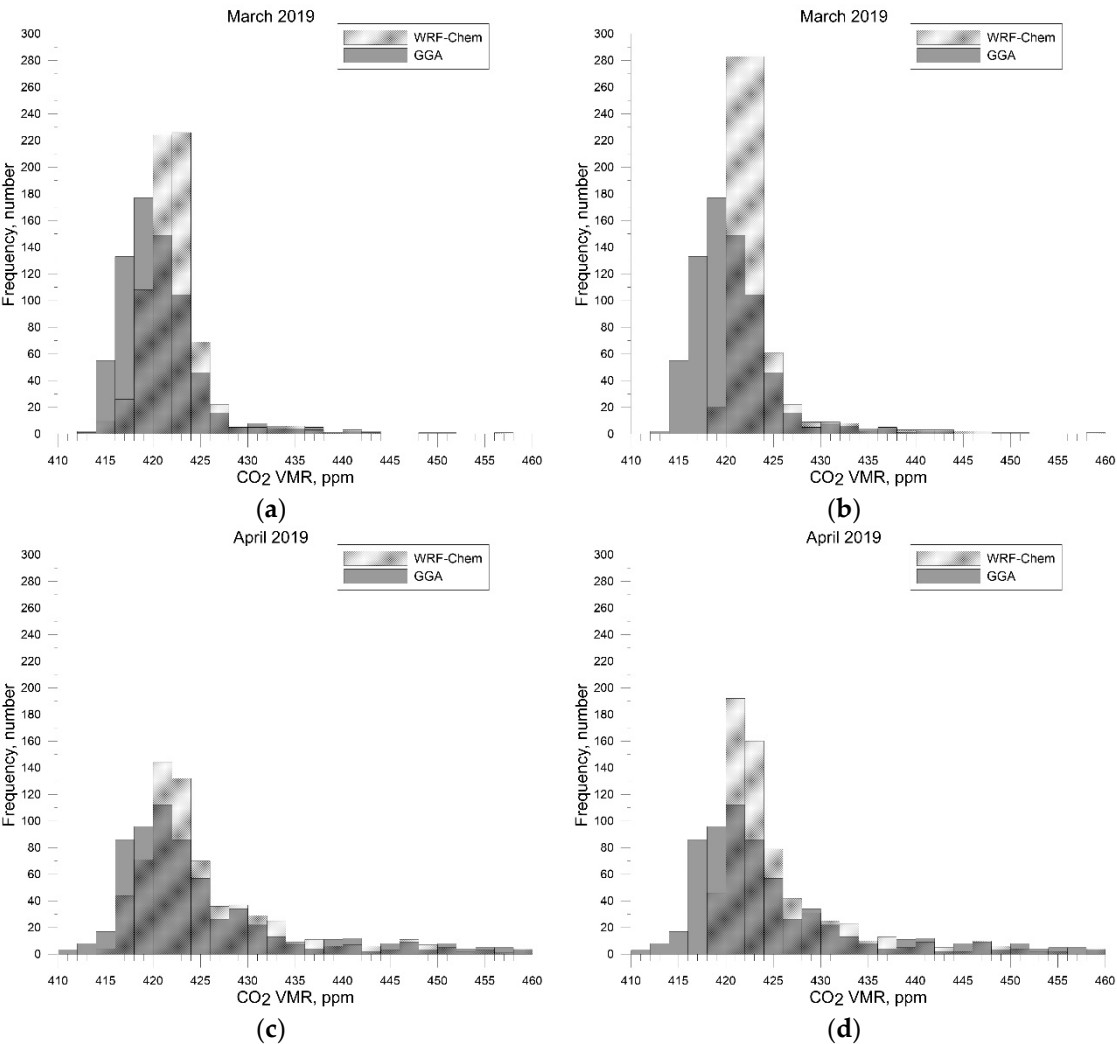

**Figure 8.** Histograms of the ground-level $CO_2$ mixing ratio according to the in situ observations and WRF-Chem data (runs with time-varying biogenic and anthropogenic fluxes (**a,c**) and with only time-varying anthropogenic emissions (**b,d**)) in Peterhof in March and April 2019.

It can be clearly seen that the means of the modelled and observation data are quite similar, especially in April, when the WRF-Chem data means were lower on approximately 1 ppm than the observation mean. In March, the modelled average mixing ratio means were 2–3 ppm higher than the observation mean. The SD values of the modelled and observation data are almost the same in March but differ by about 5–6 ppm in April, with 14.3 ppm for the observation and 8–9 for the WRF-Chem data. The larger differences in April 2019 could be caused by local processes, which hardly can be considered in such simulation and by the more complex spatio-temporal distribution of the actual $CO_2$ sources and sinks. The means and SDs of the three WRF-Chem model datasets are almost identical and differ by about 0.2 and 0.4–1.1 ppm, respectively, in both months. In general, the increase in the mean value in April 2019 was simulated well by the model. Notably, we have already provided some guesses concerning the more homogeneous trend of the $CO_2$ mixing ratio in March and the higher mean mixing ratio in April 2019. The judgements were based on the data on the spatial distribution of $CO_2$ anthropogenic emissions and the monthly averaged wind speed in Saint Petersburg in March and April 2019 (see Figure 2). Comparing the WRF-Chem data with the time-varying and constant anthropogenic emissions, it can be said that there are insignificant differences in both months. However, the mean and SD for the WRF-Chem data with the time constant anthropogenic emissions are a bit higher during both months (by 0.2–0.8 ppm). Hence, the time variance makes the WRF-Chem data match the observation means and SDs better in March and worse in April 2019.

Since the time series of the ground-level atmospheric $CO_2$ mixing ratio according to the WRF-Chem model runs where only varying (Table 1, 2ab) and constant (Table 1, 3ab) in time anthropogenic emissions were used look quite similar, in Figure 9, we present the temporal variation in the in situ observations and the data of the only two WRF-Chem model runs, with varying in time biogenic and anthropogenic fluxes (Table 1, 1ab) and with varying in time anthropogenic emissions. The differences between the observations and model data are given in Figure 10. The figures will help us to understand and illustrate how the modelled trend of the $CO_2$ mixing ratio coincides and disagrees with the real trend in particular days of the March–April 2019 period.

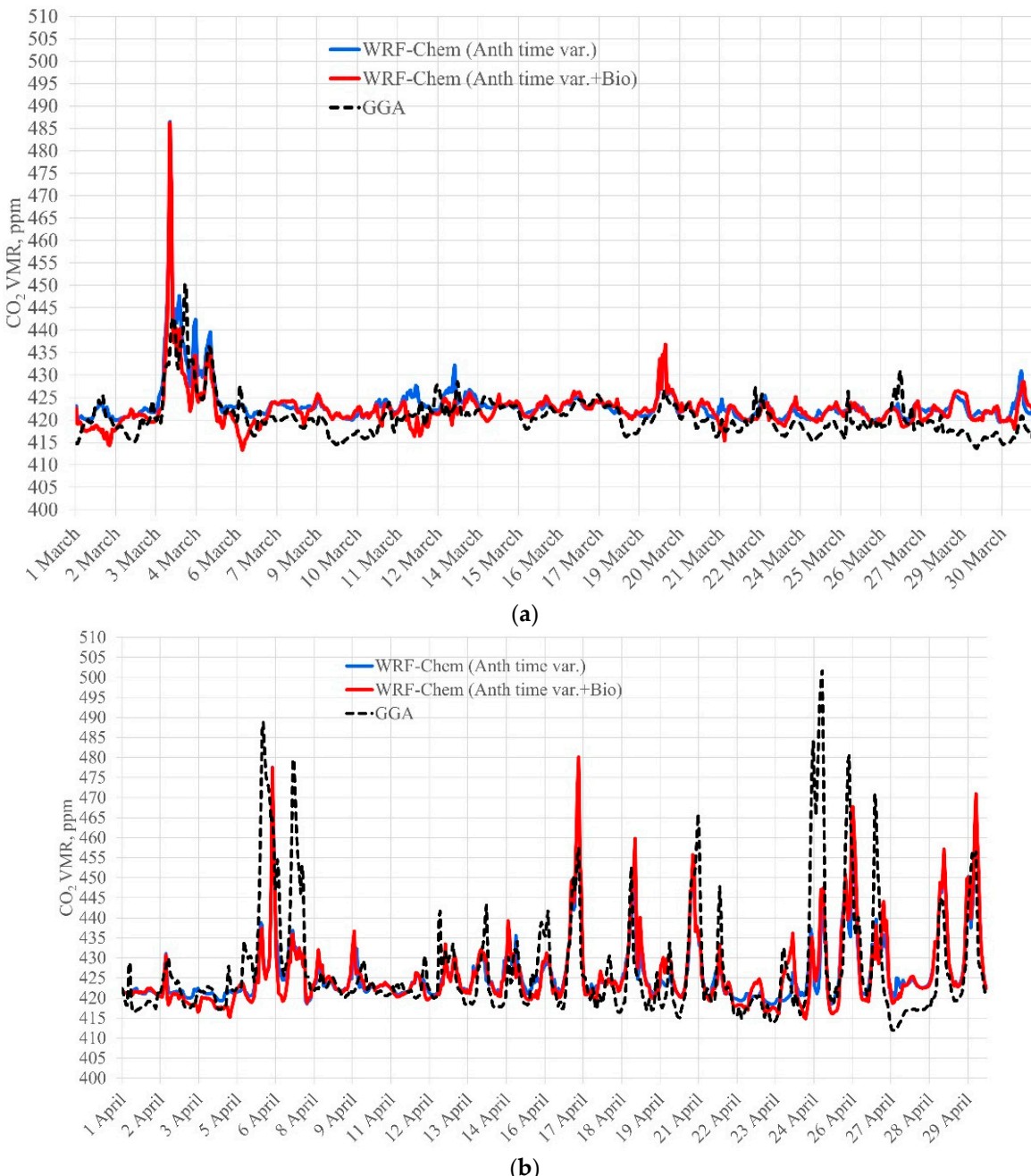

(**a**)

(**b**)

**Figure 9.** Temporal variation (1 h) in surface atmospheric $CO_2$ mixing ratio according to the WRF-Chem runs with time-varying biogenic and anthropogenic fluxes and only with time-varying anthropogenic emissions and in situ observations in Peterhof in March (**a**) and April (**b**) 2019; Anth: anthropogenic emissions; Bio: biogenic fluxes; time var.: time-varying.

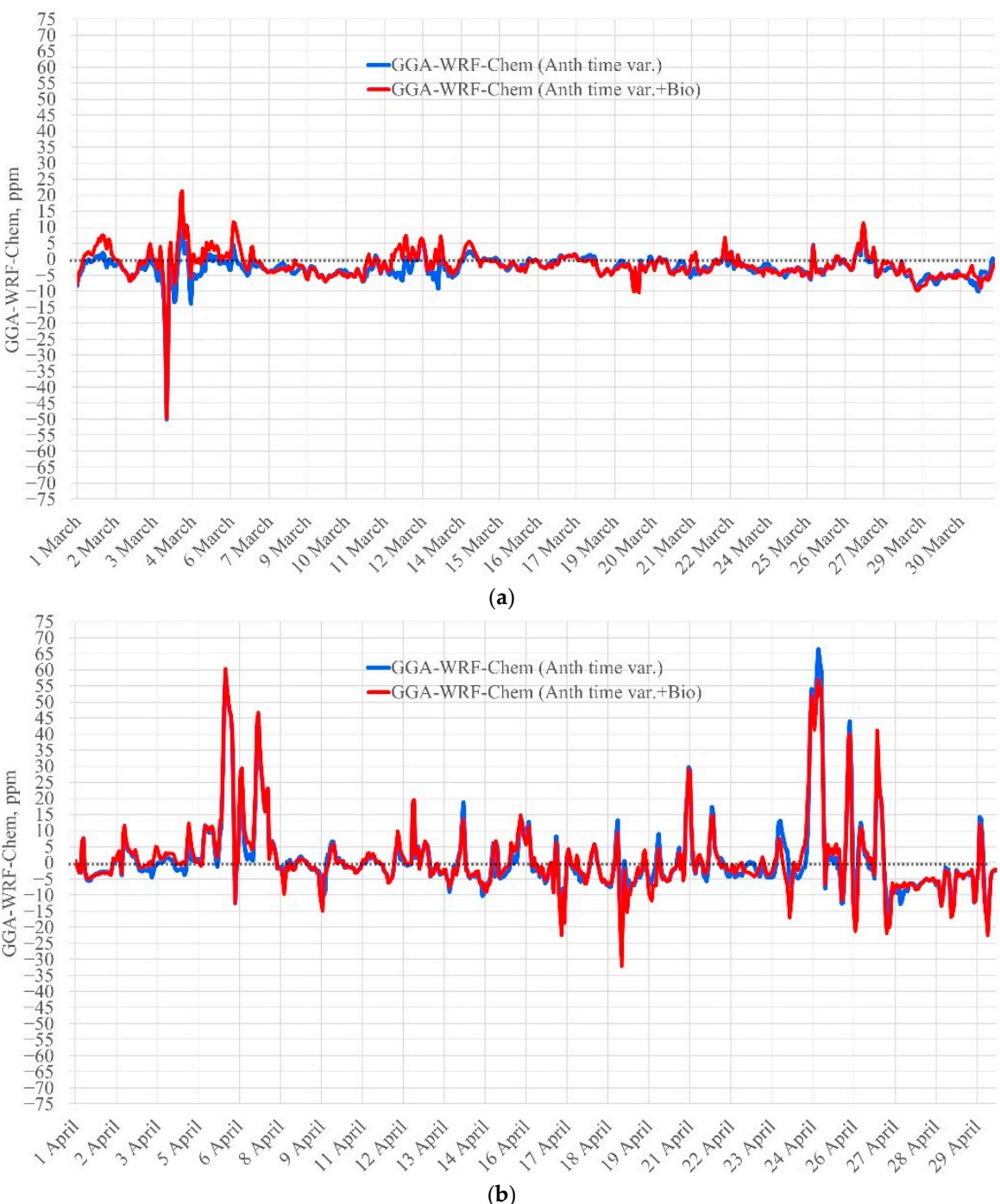

**Figure 10.** Temporal variation (1 h) in surface atmospheric $CO_2$ mixing ratio differences between the WRF-Chem runs with time-varying biogenic and anthropogenic fluxes and only with time-varying anthropogenic emissions and in situ observations in Peterhof in March (**a**) and April (**b**) 2019. Anth: anthropogenic emissions; Bio: biogenic fluxes; time var.: time-varying.

The graphs demonstrate that, in general, the WRF-Chem model simulates the temporal variation in the ground-level atmospheric $CO_2$ mixing ratio relatively well for both months. However, on some days, the mixing ratio according to the modelled data was significantly lower or higher than the observations. For instance, such biases can be seen during periods from 3 to 5 March and from 16 to 19 April where the modelled $CO_2$ mixing ratio appeared to be higher with the maximum difference on 3 March ($\approx$50 ppm). By contrast, the WRF-

Chem atmospheric $CO_2$ mixing ratio was significantly lower during the periods of 5–8 and 24–27 April with the maximum difference on 25 April ($\approx$50 ppm).

Some features of the mixing ratio trend in March and April 2019 in Peterhof can be explained by the comparison of the temporal variation in the wind characteristics (Figures 5 and 6) with the $CO_2$ mixing ratio (Figure 9). It is clear that the steep increase in the $CO_2$ mixing ratio in the period of 3–5 March (Figure 9a) matches with the wind direction, which made air masses moving from the side of Saint Petersburg urban area (Figure 6 left, approximately from 35° to 135°). The following homogeneous variation in the mixing ratio in March also can be related to the wind direction which varied from 135° to 360° (opposite to Saint Petersburg center). In contrast, the inhomogeneous temporal distribution of the $CO_2$ mixing ratio in April (Figure 9b) can be connected with the wind directions (Figure 6 right), which on most of the days changed frequently, often having values in the range of 35–135° (in more than 30% of cases). For instance, the almost east wind direction ($\approx$90°) in the period 25–30 April could cause peaks in $CO_2$ mixing ratio (see Figure 9b) according to the observation and WRF-Chem data. The discrepancies in the mixing ratio according to the WRF-Chem and observation data in the period 9–22 April could originate because of the inaccuracies in the WRF-Chem wind direction modelling (see Figure 6, right). Finally, the lower wind speed in April could also contribute to the less homogeneous trend of the $CO_2$ mixing ratio variation and the higher values of the mixing ratio in this month. Figure 10 shows that the WRF-Chem model data with the biogenic and anthropogenic fluxes included match to the observations a bit better than the WRF-Chem data with the time-varying anthropogenic emission considered (especially in April).

The comparison of monthly-averaged diurnal variation of near-surface $CO_2$ mixing ratio according to the modelled data demonstrates small but visually notable discrepancies between the WRF-Chem data with constant and varying in time anthropogenic emissions (Figure 11a,b). The largest differences were observed in April 2019 on average in the first part of a day reaching approximately 1.5 ppm (Figure 11b). In contrast, the differences in March were negligibly small (Figure 11a), but the largest discrepancies were also registered during the first 7–8 h of the day. The mean biases between two modelled datasets were 0.1 and 0.4 ppm in March and April, respectively. Additionally, the figures represent adequate agreement between the WRF-Chem modelled data and local observations in Peterhof. The best fit was registered for the WRF-Chem data with biogenic and anthropogenic $CO_2$ fluxes (mean biases constitute 1.7 and 2.1 in March and April 2019, respectively). The modelled data where only anthropogenic emissions were included on average differed from the observations on more than 2.5 ppm in both months.

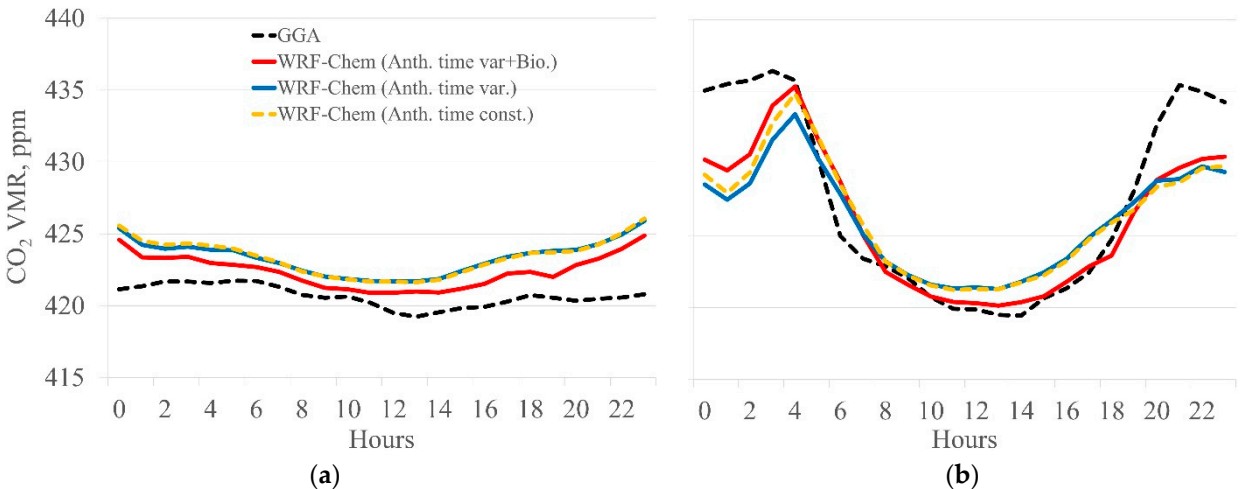

**Figure 11.** Monthly-averaged diurnal variation of near-surface $CO_2$ mixing ratio according to the WRF-Chem runs and in situ observations in Peterhof in March (**a**) and April (**b**) 2019.

Figure 11b highlights the notable diurnal cycle of the near-surface $CO_2$ mixing ratio in April 2019 according to the three WRF-Chem datasets and observations. The diurnal mixing ratio varied up to approximately 2–3 ppm in March and 15 ppm in April. We assume that such large difference in the diurnal cycles was related to the biospheric influence in April. Figure 4 depicts how the modelled and observed biospheric $CO_2$ fluxes from the territory of Hyytiälä station (southern Finland) decreased from 3 to 12 UTC with the following rise to 18 UTC. The similar trend can be seen on Figure 11b. However, in two WRF-Chem model runs, biospheric sources and sinks were not considered explicitly (blue solid and orange dotted lines in Figure 11). Perhaps, the biospheric effect was implicitly provided by the chemical boundary conditions. Probably such an influence can be significant especially in the beginning of the growing season when local biospheric $CO_2$ sources and sinks have no strong impact on near-surface $CO_2$ mixing ratio.

The next step was the assessment of three statistical characteristics, which describe the difference between the modelled and observation data, as shown in Table 7. The observed atmospheric $CO_2$ on average is 1.7–2.7 ppm lower than the simulated data in March, but approximately 1–1.3 ppm higher in April. The minimal value of RMSD was found in March (4.6 ppm) when maximal RMSD was registered in April (more than 11.6 ppm) and during March–April 2019 (8.8 ppm). Moreover, the larger mismatches in April could be caused by the discrepancies in the wind direction modelled data (M = −8.5°, RMSD = 35.1° relative to the ERA5 reanalysis). R is almost the same for three modelled datasets in April ($0.60 \pm 0.06$). The given results show that there are no significant differences between the WRF-Chem model runs data, especially in April, when the differences between Ms and RMSDs are less than 0.5 ppm and during the March–April 2019 period. However, it can be seen that there are discrepancies in the statistical parameters in March, when the M of the model run where only anthropogenic emissions were used is on 1 ppm higher than the M of the WRF-Chem run with biogenic and anthropogenic fluxes considered. By contrast, in March, the maximal R was found for the modelled data where only anthropogenic emissions were used ($0.70 \pm 0.05$). The WRF-Chem data where biogenic and anthropogenic $CO_2$ fluxes were considered have a minimal R ($0.55 \pm 0.06$).

**Table 7.** Statistical characteristics of difference between the WRF-Chem and observation data for Peterhof in March and April 2019. Here, t.v. is temporal variation and t.conss. is time constant.

| Period | March 2019 | | | April 2019 | | | March–April 2019 | | |
|---|---|---|---|---|---|---|---|---|---|
| GGA− WRF-Chem | t.v. Anth + Bio | t.v. Anth | Const. Anth | t.v. Anth + Bio | t.v. Anth | Const. Anth | t.v. Anth + Bio | t.v. Anth | Const. Anth |
| M, ppm | −1.7 | −2.7 | −2.7 | 1.3 | 1.2 | 1.0 | −0.3 | −0.8 | −0.9 |
| RMSD, ppm | 4.7 | 4.6 | 4.6 | 11.5 | 11.4 | 11.6 | 8.7 | 8.6 | 8.8 |
| R | $0.55 \pm 0.06$ | $0.69 \pm 0.05$ | $0.70 \pm 0.05$ | $0.60 \pm 0.06$ | $0.60 \pm 0.06$ | $0.58 \pm 0.06$ | $0.61 \pm 0.04$ | $0.62 \pm 0.04$ | $0.61 \pm 0.04$ |

The analysis depicts that the temporal distribution of the near-surface $CO_2$ mixing ratio in Peterhof in March and April 2019 according to the WRF-Chem simulations where three different sets of the fluxes were used look almost similar. A little more differences between the WRF-Chem data can be seen for the particular month. For instance, distinct discrepancies were registered between the modelled data with and without the biogenic $CO_2$ emissions for the mean biases (differ on ≈ 1 ppm), correlation coefficients (0.55 vs. $0.70 \pm 0.06$, respectively) and mean concentrations (differ on ≈ 1 ppm) in March. The modelling with the anthropogenic and biogenic fluxes considered gave the best fit to the observation data, according to the graphs of the WRF-Chem $CO_2$ time series. The diurnal variance applied to the anthropogenic emissions influenced the WRF-Chem data insignificantly in comparison to using the biogenic fluxes. Perhaps this is due to the simplicity of the time variance used, which did not consider the city impact on the anthropogenic emissions. In addition, the influence of the diurnal anthropogenic emission variance

could be significantly smaller than the impacts of meteorological state or the WRF-Chem boundary conditions.

However, the biogenic fluxes did not change the WRF-Chem data significantly relative to the other WRF-Chem data, considering the full March–April 2019 period. The most obvious reason for this could be the late beginning of the growing season (at the end of April or the beginning of May 2019), which determines the start and the end of the activity of $CO_2$ biogenic sources and sinks. That could lead to the small values of real biogenic fluxes or even to the absence of the fluxes. The second reason for this could be the errors in the biogenic emission estimation.

## 6. Conclusions

Inversion modelling is a relatively new and attractive technique that combines different measurements and the data of chemical transport models (CTMs) to estimate $CO_2$ emissions at various scales. Many studies have demonstrated that the estimations are sensitive to the quality of CTMs (especially at a high spatial resolution) and a priori information, which includes the spatio-temporal variation in different fluxes, meteorological data, and the chemical initial and boundary conditions. In our research, we provided a quality analysis of the a priori information (CAMS data) and WRF-Chem data for the temporal variation in the near-surface $CO_2$ mixing ratio and the wind at 10 m in a suburb of St. Petersburg in March and April 2019. We validated the WRF-Chem wind data using ERA5 meteorological reanalysis when the CAMS and WRF-Chem data on the near-surface $CO_2$ mixing ratio were compared to the in situ observations in Peterhof.

1.  In general, the WRF-Chem model is able to simulate the wind speed and direction at 10 m in the suburb of Saint Petersburg quite accurately with respect to the ERA5 meteorological reanalysis. However, the analysis for the particular months demonstrates that the wind directions between the two datasets disagree more in April 2019 (RMSDs $\approx 35°$). The wind direction discrepancies in March are approximately two times lower. We suppose that the differences between the WRF-Chem and ERA5 wind parameters in Peterhof in March and April 2019 could be related to the meteorological boundary conditions used in the WRF-Chem simulation and the specific meteorological situation in April 2019, which could not be represented by the model reasonably well.
2.  Different CAMS products can be employed as a priori data in the inverse modelling of $CO_2$ emissions. The comparison of the CAMS products (reanalysis and analysis) with each other and the observation data for the near-surface atmospheric $CO_2$ mixing ratio in Saint Petersburg in March and April 2019 demonstrate the high variability of the differences depending on the month. Despite the fact that the spatial resolution of the CAMS reanalysis is notably lower than the analysis, the first one fits the observations better than the last one in March 2019. The analysis data overestimate the real $CO_2$ mixing ratio significantly (on average by more than 10 ppm). The large discrepancies between the analysis and observations can be related to the estimation errors of the $CO_2$ fluxes for the non-urbanized territories used in the CAMS modelling. For example, the CAMS analysis matches the observed $CO_2$ mixing ratio trend in Peterhof relatively well with wind from the Saint Petersburg urbanized area in April 2019. These agreements are related to the high spatial resolution of the CAMS analysis data ($\approx$15 km). By contrast, the CAMS reanalysis data agree with the observations in April significantly worse than in March. These differences can be caused by the low spatial resolution of the CAMS reanalysis ($\approx$200–300 km), which makes it impossible to detect the influence of the Saint Petersburg urbanized area on the $CO_2$ mixing ratio in Peterhof.
3.  In general, the regional numerical weather prediction and chemistry transport model WRF-Chem adequately simulates the temporal variation in the near-surface $CO_2$ mixing ratio with a high spatial resolution (3 km) in Peterhof (Saint Petersburg) in March and April 2019. It is worth noting that despite the acceptable accuracy of

the WRF-Chem data, the CAMS analysis used as the chemical initial and boundary conditions overestimates the observation data significantly—the mean biases and RMSDs are in ranges from $-12$ to $-14$ ppm and 14 to 19 ppm, respectively. Besides this, the estimation errors for the biogenic fluxes and anthropogenic emissions could also contribute to the mismatches between the observation and WRF-Chem data. Finally, applying the wrong or simple time variance to the anthropogenic $CO_2$ emissions could have caused the discrepancies in the measurements. However, to verify these assumptions, extra WRF-Chem modelling without anthropogenic emissions is needed.

4.　The diurnal variation in the $CO_2$ anthropogenic emissions influenced the WRF-Chem data insignificantly in comparison to including the biogenic fluxes in the simulation. It was shown that the biogenic fluxes caused the WRF-Chem data to fit the in situ observations in Peterhof in March and April 2019 a bit better. Perhaps the diurnal variation effect was negligible due to its simplicity or incompleteness. In fact, this variation does not take into account the variability in weekly anthropogenic emissions. The analysis of the monthly-averaged diurnal cycle of near-surface $CO_2$ mixing ratio represented that the diurnal variation of the anthropogenic emissions caused small but visually notable difference between the modelled data in April. The average diurnal cycle, according to the WRF-Chem data with the biogenic fluxes included, fitted the observations a little better. We assume that the relatively small effect of the biogenic fluxes on the WRF-Chem data can be connected to the late beginning of the growing season (e.g., the end of April 2019), which influences the $CO_2$ transfer between the atmosphere and vegetation. To confirm this, we plan to provide WRF-Chem simulations for periods with contrasting biogenic activity (e.g., winter and summer months) for the same area. Besides this, we would like to use online VPRM, which is part of the current WRF-Chem release. We suppose that in the beginning of the growing season, the chemical boundary conditions can influence near-surface $CO_2$ mixing ratio more significant than the biogenic sources and sinks considered explicitly within the modelling domain. Therefore, in our further research, we would like to study the role of chemical boundary conditions in the simulation of the ground-level $CO_2$ mixing ratio especially in the beginning and middle of the growing season.

5.　The wind direction variations were essential for the temporal distribution of the near-surface $CO_2$ mixing ratio in Peterhof in March and April 2019. We demonstrated that the wind from the Saint Petersburg urban area led to a significant increase in the Peterhof $CO_2$ mixing ratio and to the inhomogeneous trend of the mixing ratio variation. In contrast, it was shown how the opposite wind caused a decrease in the mean mixing ratio and led to the homogeneous $CO_2$ mixing ratio temporal variation. According to the analysis, the WRF-Chem model adequately simulates the wind speed and direction (except for some days in April 2019). Therefore, when providing WRF-Chem simulations for Saint Petersburg with a spatial resolution of 3 km, more attention should be given to the quality of the chemical initial and boundary conditions and $CO_2$ sources and sinks.

To sum up, the main factors that determined the near-surface $CO_2$ mixing ratio in Peterhof in March and April 2019 were the meteorological conditions (wind speed and direction in the surface layer) and anthropogenic $CO_2$ emissions from the Saint Petersburg urbanized area. Nevertheless, the $CO_2$ transport from the territories outside the area of interest could also influence the $CO_2$ mixing ratio in Peterhof significantly. In this study, we demonstrated that the high-resolution numerical model WRF-Chem can simulate the surface-layer $CO_2$ mixing ratio in Peterhof relatively well. However, to investigate whether the model is suitable for the inverse modelling of the $CO_2$ anthropogenic emissions from the territory of Saint Petersburg, further analysis is needed (in particular, an accuracy analysis of the simulation of the $CO_2$ total column content), which we are planning to carry out in the next study.

**Author Contributions:** Conceptualization, G.N., Y.T. and S.S.; data curation, S.F. and I.M.; formal analysis, G.N., Y.T., S.S. and Y.V.; funding acquisition, S.S.; investigation, G.N., Y.T., S.S. and Y.V.; methodology, G.N., Y.T. and S.S.; project administration, G.N.; resources, S.S.; software, G.N.; model set-up and run, G.N.; validation, G.N., Y.T., S.S. and Y.V.; visualization, G.N. and Y.V.; writing—original draft, G.N., Y.T. and S.S.; writing—review and editing, G.N., Y.T., S.S., S.F., I.M. and Y.V. All authors have read and agreed to the published version of the manuscript.

**Funding:** Adaptation of the WRF-Chem model to the Saint Petersburg region and numerical calculations were carried out on the RSHU computing cluster with the support of the state task of the ministry of science and higher education of the Russian Federation (project no. FSZU-2020-0009).

**Institutional Review Board Statement:** Not applicable.

**Informed Consent Statement:** Not applicable.

**Data Availability Statement:** The GFS data used in the WRF-Chem simulations are openly available at https://www.ncdc.noaa.gov/data-access/model-data/model-datasets/global-forcast-system-gfs (accessed on 19 December 2019) (GFS-ANL). The CAMS analysis data are available on request from copernicus-support@ecmwf.int. The CAMS reanalysis data are openly available at https://ads.atmosphere.copernicus.eu/cdsapp#!/dataset/cams-global-greenhouse-gas-inversion?tab=form (accessed on 14 May 2020). The ODIAC fossil fuel emission dataset used in the WRF-Chem simulations is openly available at http://db.cger.nies.go.jp/dataset/ODIAC/DL_odiac2019.html (accessed on 13 December 2019), DOI: doi:10.17595/20170411.001. The VPRM model source code and description are available at https://github.com/Georgy-Nerobelov/VPRM-code (accessed on 13 November 2020). MODIS reflectance data are openly available at https://lpdaac.usgs.gov/products/mod09a1v006/ (accessed on 15 November 2019). MODIS land cover type data are openly available at https://lpdaac.usgs.gov/products/mcd12q1v006/ (accessed on 15 November 2019). ERA5 wind, air temperature, and incoming short-wave radiation data are openly available at https://cds.climate.copernicus.eu/cdsapp#!/dataset/reanalysis-era5-single-levels?tab=form (accessed on 26 December 2020). The "mozbc" and "anthro_emiss" tools used in the WRF-Chem simulations are openly available at https://www.acom.ucar.edu/wrf-chem/download.shtml (accessed on 10 October 2019). The WRF-Chem and VPRM simulation data are available on request (please contact akulishe95@mail.ru; akulishe95@gmail.com).

**Acknowledgments:** We acknowledge use of the WRF-Chem preprocessor tools "mozbc" and "anthro_emiss" provided by the Atmospheric Chemistry Observations and Modeling Lab (ACOM) of NCAR. Additionally, we would like to thank the CAMS for providing the free data of the analysis of $CO_2$ spatio-temporal distribution on global scale and authors of the ODIAC dataset. We are thankful to the ECMWF for the ERA5 reanalysis data. In addition, we would like to express our sincere gratitude to Maria Makarova and Anatoly Poberovsky who provided the in situ observation data of atmospheric $CO_2$ mixing ratio in Peterhof. The authors are thankful to the IT group of the RSHU computer cluster and to Maxim Motsakov (RSHU) for his valuable advises on the adaptation of the WRF-Chem model. The Peterhof observations were carried out using the scientific instruments of the "Geomodel" research center of Saint Petersburg State University.

**Conflicts of Interest:** The authors declare no conflict of interest.

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
