# Peer review of "Validation of WRF-Chem Model and CAMS Performance in Estimating Near-Surface Atmospheric CO2 Mixing Ratio in the Area of Saint Petersburg (Russia)"

_atmosphere, doi:10.3390/atmos12030387_

Round 1

Reviewer 1 Report

I would like to thank the authors for their careful revisions and responses to the review’s initial comments. The revised manuscript has been improved substantially. However, I still have a few comments that hopefully can be addressed during the second-round revision before acceptance for publication.

Firstly, as the authors presented a model study of CO2 variations at a suburban site Peterhof near the city of St. Petersburg, the introduction should set the context for the city-scale CO2 observations and simulations. Why city-scale greenhouse gas studies are important? What are the main challenges of CO2 transport modeling and inverse modeling at this scale? Are there any previous studies working on this topic (for example, using WRF-Chem for CO2 transport modeling at a city scale)? The references cited in the introduction are more related to large scale CO2 transport modeling and inversions (e.g., ref. 22 that the authors highlighted in Line 85-92). Besides, the mention of satellite observations (Line 68-75, Line 82-83) is not quite relevant, although some of them can be useful for city-scale GHG emission estimation (OCO2 for example). For one thing, you study didn’t use any satellite observations or provide any implications for satellite-based estimations of GHG emissions; on the other hand, the inverse problem in the context of your study is more related to the inversion of atmospheric transport rather than inversion of atmospheric optics that is used to obtain satellite GHG measurements (Line 72-73).

Secondly, I notice that the authors have added a section to compare the meteorology (wind fields in particular) simulated by WRF-Chem to ERA5 (Section 5.1). I am wondering if there are any meteorology stations in Peterhof or St. Petersburg that you can compare to (there should be). Note that the horizontal resolution of ERA5 is ~30 km, much coarser than those for WRF-Chem (9 km and 3 km for the outer and inner domains, respectively). Therefore, it’s still not clear whether WRF-Chem can simulate well the fine-scale meteorology, even though WRF-Chem and ERA5 agree well with each other at the gridcell where Peterhof is located. I would suggest the authors compare both WRF-Chem and ERA5 meteorology to observations at Peterhof and examine whether the performance of WRF-Chem is better than that of ERA5 due to its finer resolution.

Thirdly, I still don’t quite understand why the authors compared CAMS CO2 reanalysis to observations at Peterhof. If this is done for the use of CAMS reanalysis in CO2 inversions in the future, as the authors argued in their response to my general comment #6, why did you perform your WRF-Chem transport modelling prescribed with ODIAC anthropogenic emissions and VPRM biogenic fluxes? If this is done to show that the CAMS reanalysis better matches observations at Peterhof than CAMS analysis (at least for March) due to assimilation of CO2 observations, why didn’t you use CAMS reanalysis for boundary conditions? I would suggest removal of the comparison for CAMS reanalysis from your study. For CAMS analysis, if you want to justify its use as boundary conditions, indeed it would be more helpful if you could compare the simulations with observations outside your nested domains (global or regional background stations in Europe not far from your domain, e.g. Mace Head (MHD) in Ireland, Pallas-Sammaltunturi (PAL) in Finland, for site location and data download please check in https://www.esrl.noaa.gov/gmd/dv/site/).

Fourthly, for CO2 transport modeling (and inversion) focusing on urban areas, it’s very important that the short-term variations can be sufficiently resolved in the transport model. Fig. 9 showed that there were occasions where peaks were substantially overestimated or underestimated. These could be good opportunities for case studies – whether the mismatch could be due to inaccurate anthropogenic emissions or meteorology (e.g. wind fields, boundary layer height, etc). The observations of meteorology at Peterhof (if there’s any) and the evaluation of simulated meteorology by WRF-Chem suggested in my second comment would be helpful for error attribution. Besides, the comparison of monthly averaged diurnal cycles could be also interesting to look at. The authors mentioned in their revision that the simulated time series of experiment 2 (with diurnal profile of anthropogenic emissions) and experiment 3 (with constant anthropogenic emissions) looked rather similar and therefore the latter one was not presented in the manuscript. If the authors compare the simulated diurnal cycles with the observations, we would expect differences between the two experiments.

Lastly, I feel the results and discussions for model evaluation are too long and tedious to read. The authors should try to condense the key messages of figures and tables and organize them in a logic way. The comparison of CO2 frequency distribution in Fig. 8 does not contain much useful information in terms of model evaluation, and the authors may consider to remove it. Apart from tables that showing mean bias, RMSD and R between model simulations and observations, I would suggest the authors to use Taylor diagram to visualize model-observation discrepancies (see https://agupubs.onlinelibrary.wiley.com/doi/abs/10.1029/2000JD900719 for details). In the conclusions, the authors should also address the implications of the study in addition to the summary of results. For example, is the WRF-Chem transport model with current configurations capable for CO2 inversions? What are the model strength and weakness for city-scale transport modeling? Based on your study, do you have any suggestions for model improvement in the next steps?

Specific comments and technical corrections:

Line 62-64: Any citations here?

Line 142-144: What are the dominant wind directions during March and April 2019? I understand they are presented in Figure 2, but not so clear in the main text. And how about the distribution of wind speed? The authors may consider to plot wind roses, which would be nice to show distribution of both wind directions and speeds at the same time (see for example Fig. 3 in Xueref-Remy et al. (2018) https://acp.copernicus.org/articles/18/3335/2018/).

Line 153-155: I don’t quite understand why the average CO2 mixing ratio in March should be lower than that in April?

Line 156-162: Please locate Peterhof stations on the maps.

Line 217: weekly -> weakly

Line 222: “emitted and consumed” – when you talk about vegetation, it would be more relevant to say they “release and absorb” CO2.

Line 327-328: RMSD not only characterizes large outliers in differences, but discrepancies of two datasets in general (imagine that RMSD could be also large for two CO2 time series with a constant difference of e.g. 10 ppm).

Reviewer 2 Report

I am fine with the revised manuscript except that I would like the authors to incorporate their reply to my comments #2 and #3 in the revised manuscript, in order to address my original comment #1: "The technical details need substantial improvements".

Reviewer 3 Report

The authors have done a great job in addressing the reviewer comments – this manuscript is now much easier to follow and provides clear conclusions that will be of interest to the wider community. I found the additional comparison of the two CAMS products particularly interesting. I have a couple of additional points that I think deserve further attention – once these are addressed I suggest this paper is suitable for publication.

I missed this last time, but looking at Figure 4 again I’m slightly confused as to why these particular model cells have been chosen for comparison. Are the coordinates given in the legend the cell centre coordinates? If so, then I guess the flux tower is in the cell shown by the green line? If this is true I think it is worth stating it explicitly, in addition to an explanation of how the other cells were chosen. Really at this resolution (0.1 degrees) I’m not sure one would expect good agreement with a flux tower, whose measurements represent fluxes at the local scale. So I don’t think the data presented here are any cause for concern, but equally the fact that the flux tower measurements agree with the VPRM fluxes in a seemingly random grid cell (blue line) does not provide additional confidence in the VPRM fluxes! I would replace the sentence in L247 with a statement saying that you don’t expect perfect agreement between local-scale flux tower measurements and the coarse resolution model data, but that the diurnal cycle in the model closely resembles the measurements. Also, unless I missed it, I don’t think the resolution of the VPRM run is stated anywhere. If not then this should be added to this section.

Figure 6 is hard to interpret because there are large spikes where the wind directions cross between 360 and 0 degrees. The maximum difference in wind direction is really 180 degrees – any differences larger than this should be corrected by subtracting the difference from 360. For example, a difference of 355 degrees is really a difference of 5 degrees. I think the same issue affects the values in Table 3.

L489-490 – What is meant by “the first and the third decades of April”? Maybe just give dates?

Round 2

Reviewer 1 Report

The authors have addressed most of my questions. Thanks. For my major comment #4, as I said, the model's capability to reproduce short-term variability is important for city-scale studies. I would like to see results and discussions on the model-observation comparisons of monthly-averaged diurnal cycles. I understand that the authors did the analyses (as they claimed in their responses). It would be nice that these results could be shown in the manuscript.

Author Response

This manuscript is a resubmission of an earlier submission. The following is a list of the peer review reports and author responses from that submission.

Round 1

Reviewer 1 Report

This paper evaluated the performance of a high-resolution transport model WRF-Chem in simulating CO2 variations at a suburban site Peterhof near St. Petersburg during March and April, 2019. Model simulations were compared to observations with various statistics, including monthly mean, maximum, minimum, standard deviations, etc. Sensitivity tests were performed to show capabilities and limitations of WRF-Chem to represent short-term CO2 variations, particularly for the accuracy of biogenic CO2 fluxes from VPRM. The data product of CAMS CO2 posterior concentrations (based on fluxes from a global inversion at a much coarser resolution) was also compared to observations at Peterhof and showed larger model-obs discrepancies than WRF-Chem did, which according to the paper may suggest the advantage of high-resolution transport model in capturing the short-term variabilities at a local scale.

While the authors provided a case study of CO2 transport model simulation and evaluation at a suburban site, the key messages of the paper are not clear. The abstract and conclusion are rather descriptive than conclusive. Is WRF-Chem capable to simulate short-term variabilities at local scale and suitable for regional CO2 inverse modeling? If not, which aspects of the transport model should be improved? Why model simulations without biogenic fluxes performed better? Was it due to errors in local-scale meteorology or inaccurate modeling of biogenic fluxes by VPRM? The authors should able to answer these questions in this paper and distill the key messages behind the data and results.

I also find the design of the study and the ways that the authors analyzed, interpreted and presented the data were questionable. Here are a few of my comments and suggestions:

  • In order to reproduce the CO2 variabilities at high spatial and temporal resolutions, one of the prerequisites is the accuracy of meteorology. Is WRF-Chem able to simulate the meso-scale or local scale meteorology during the study period? Does it agree well with observations or meteorological reanalysis datasets? I didn’t see any comparisons or discussions in the paper.
  • I’d like to see a map showing the region where the suburban site Peterhof is located, with the spatial distribution of major anthropogenic sources, vegetation, CO2 emissions from ODIAC, and the dominant wind directions during the study period. There should also be a paragraph in Method accordingly that describes the area around the measurement site, its location with respect to the megacity St. Petersburg and major anthropogenic sources, wind directions etc. This would be very helpful for interpretation of the peaks shown in Figure 3 and origins of sources.
  • The authors used various metrics for model-observation comparison, including the monthly means, max, min, mean bias, RMSE, etc. I am not sure whether these metrics are useful and meaningful to assess the model’s ability in reproducing CO2 variabilities. For example, the model-observation comparisons of RMSE, max, and min (Table 2 & 4) do not make much sense, as two completely different time series could give rather similar values of these metrics (imagine two time series with the same max/min but different timing of max/min). The comparisons of monthly means (Table 2 & 4), as well as the calculation of mean biases and RMSE (Table 3 & 5), which were based on absolute CO2 concentrations, are also questionable. The model-observation discrepancies of absolute CO2 concentrations could depend on boundary conditions such as the initial state of model simulations. You may consider looking into whether the model can capture the CO2 enhancement (e.g., the peaks) in reference to the background. To do this you may need to define a background concentration representative of clean air mass from a particular wind sector. Figure 3 shows that the model can well capture the occurrence of CO2 peaks, but may overestimate or underestimate the magnitude, which could be due to errors in local meteorology, inaccurate source distribution or emission intensity, inaccurate emission scale factors (you accounted for the diurnal cycle of CO2 emissions, but you didn’t take into account of differences between weekdays and weekends?), etc.
  • I don’t quite understand why the authors evaluate the CAMS CO2 data product against the observations. Did you want to compare these two model outputs with different resolutions to show that WRF-Chem outperformed CAMS in simulating CO2 variations at a suburban site, as it is a high-resolution transport model? Apart from the spatial resolution, there are other factors that could contribute to the differences in model performance, e.g. the transport model per se, the quality of input fluxes and meteorology, etc.

The writing of the paper also needs substantial and careful editing and improvement, especially the introduction. It was not well structured, and sometimes off the topic completely (e.g., the paragraphs and sentences on satellite observations and CH4 or N2O transport/inverse modeling). The authors should focus on CO2 observations and transport modeling, particularly at local/city scales. Why do we need GHG observations and modeling at this fine scale? What are the challenges of high-resolution regional transport modeling? Are there any previous studies in this area? The authors should give an overview of the research topic and present the goal of your study (and the questions you’d like to answer as well) in this section.

Reviewer 2 Report

The study uses WRF-Chem and satellite-derived carbon fluxes to simulate the CO2 concentration over St Petersburg, which is then compared with the station time series and a global forecast model at the same location. The authors found that that the WRF-Chem simulated CO2 is in less agreement with the station time series if a biogenic CO2 flux is used compared with the same WRF-Chem simulation without the biogenic flux. Thus the authors suggest that the biogenic CO2 flux database may need to be improved. The global forecast model also shows significant difficulties in capturing the short-term variability over St Petersburg.

The science and the results are straightforward: Comparison of station measurement with regional and global models. A major problem of the manuscript is perhaps the writing part. The technical details need substantial improvements.

The authors discussed how they processed the station data. Are there any references about the 15-min median method? How did they arrive at the conclusion that the hourly average based on the 15-min medians characterizes CO2 on "extended air masses"?

When preparing the wind and CO2 flux data for the WRF-Chem model, how did they interpolate these data onto the model grids?

The authors only said that the ODIAC has a high resolution but they did not specify the exact resolution in their text.

Their conclusion about CAMS having the least agreement with the station data is perhaps not surprising. A more meaningful conclusion maybe what do the authors anticipate being the major factor that contributes to the disagreement? Wind fields? CO2 flux? Or simply the coarse resolution of CAMS?

The authors turned VPRM CO2 flux on and off. Did they also turn the ODIAC CO2 flux on and off as a control experiment?

Reviewer 3 Report

This study presents a comparison between in situ CO2 measurements and output from two CTMs: the CAMS global CO2 inversion product and a nested regional WRF-Chem simulation. A variety of statistics are calculated to assess the performance of both CTMs in simulating CO2 mole fractions at a suburban measurement site near St Petersburg. The authors find that the coarse resolution (3.75° longitude x 1.875° latitude) CAMS data exhibit poorer agreement with measurements than the nested WRF-Chem output (3 km resolution inner domain). This is not a particularly surprising result, but I assume that this paper is being published as a precursor to an inverse modelling study, so I understand the desire to publish analysis on which subsequent model uncertainty estimates can be based. However, I think this analysis needs some additional work before it can be published to increase its utility for this purpose. As a general point, a full English language edit would also make the paper much easier to follow.

The paper would benefit from a couple of extra WRF-Chem runs. Because the runs that include the VPRM sink term also include temporal variability in ODIAC, it is not possible to separate the impact of these two differences (relative to runs 2a and 2b). It would be good to include runs that had temporal variability in ODIAC but not the VPRM biospheric fluxes. Then this ambiguity can be removed from the conclusions section. Also, the fact that the 2b run was conducted first is not really a justification for using a different domain configuration – this should be rerun using the same configuration as the other three.

There is currently no explanation of why different CAMS datasets were used for the WRF-Chem boundary conditions than for the comparison against measured data. I assume that the inversion product was used for the comparison because the assimilation of surface measurements is expected to make it more accurate than the NRT product. And possibly the NRT product was used for the boundary conditions to provide higher spatial resolution? But given that coarse spatial resolution is one of the probable causes for the poor agreement between the CAMS inversion product and the data, it would be useful to include the higher resolution NRT data in this comparison too. Also, the boundary condition CAMS data is listed as forecast data, but I assume it was actually the analysis step (T+0) – if so then that should be clarified. If not, the forecast step that was used should be stated and an explanation given as to why this was chosen.

The results of the paper are not currently presented in a clear manner. Firstly, I’m confused by some of the RMSE values that are stated. RMSE values are given for both the measurement and modelled datasets (e.g. in Table 2). But to calculate the RMSE you need to know the truth – taken elsewhere in this paper to be the measured mole fractions – so what is the meaning of the RMSE for the measured dataset? It would be useful to have the measured RMSE relative to any target cylinder calibration data (i.e. measurements that were not included in the calculation of the gain curve), so we have a quantification of the measurement uncertainty. But that can’t be what is presented here – these values are way too high. This definitely needs clearing up – the RMSE of the model relative to the measurements (Table 3) is a useful statistic, but at the moment this whole section is very confusing. The same problem reoccurs throughout the paper (e.g. in the conclusion where it says the model “underestimates the errors in April” – what does this mean?).

The results section of the paper should be reconfigured so that the different WRF-Chem runs and the CAMS results are presented together. The comparison with CAMS is done at 3 hourly temporal resolution, and it is stated that WRF-Chem has been averaged to 3 hourly resolution to enable this comparison, but the only WRF-Chem statistics presented were calculated on an hourly basis. The section should start by comparing everything at 3 hourly temporal resolution, as far as possible combining data from the different models and scenarios into the same figures and tables so they can be directly compared. Then if the authors want to present hourly results for WRF-Chem this could be done in a subsequent sub-section.

Following on from this, the conclusions section needs expanding – currently it simply restates some of the statistics from the results section. The authors should try to address why the CAMS performance is poorer – is it the horizontal resolution? Adding a comparison with the NRT CAMS data could shed some more light on this. Similarly, while there is some brief discussion, more investigation should be undertaken to identify (or at least suggest) the reasons why adding biospheric fluxes from VPRM made the WRF-Chem simulation perform worse. This is a potentially interesting result, but at the moment it is only given a cursory mention. Again, adding runs with temporal variability in ODIAC but no biospheric fluxes would help to remove some of the ambiguity here. Currently there are no conclusions provided that are of interest to a broader audience, just some statistics that have been calculated for the specific case presented in this paper. I think there is scope to draw some more general inferences from this data which would make this study more interesting for others working in the field.

Specific points:

L25-28 – Related to the main point above, this needs rephrasing as currently it isn’t clear what is meant by this section (e.g. how can WRF-Chem “underpredict RMSE”?)

L55-57 – This range seems quite conservative, and should at least be backed up with a citation

L207 – Based on the product version number given in the text, I’m pretty sure that the CAMS inversion product used here is based on in situ measurements, not satellite data (which is used by the CAMS NRT product)

L304-307 – Perhaps I’ve misunderstood, but this seems like an example of VPRM error alluded to in the previous sentence.